:·ö·: PLOS | ONE

# An exclusive human milk diet for very low birth weight newborns—A cost-effectiveness and EVPI study for Germany

**Stefan Michael Scholz[ID]\*, Wolfgang Greiner**

Department of Health Economics and Health Management, School of Public Health, Bielefeld University, Bielefeld, Germany

\* stefan.scholz@uni-bielefeld.de

**Data Availability Statement:** All relevant data are within the manuscript and its Supporting Information files.

**Funding:** Both authors, SMS and WG, wish to disclose that the research on which our manuscript

## Abstract

### Objectives

Human milk-based fortifiers have shown a protective effect on major complications for very low birth weight newborns. The current study aimed to estimate the cost-effectiveness of an exclusive human milk diet (EHMD) compared to the current approach using cow's milk-based fortifiers in very low birth weight newborns.

### Methods

A decision tree model using the health states of necrotising enterocolitis (NEC), sepsis, NEC + sepsis and no complication was used to calculate the cost-effectiveness of an EHMD. For each health state, bronchopulmonary dysplasia (BPD), retinopathy of prematurity (RoP) and neurodevelopmental problems were included as possible complications; additionally, short-bowel syndrome (SBS) was included as a complication for surgical treatment of NEC. The model was stratified into birth weight categories. Costs for inpatient treatment and long-term consequences were considered from a third party payer perspective for the reference year 2017. Deterministic and probabilistic sensitivity analyses were performed, including a societal perspective, discounting rate and all input parameter-values.

### Results

In the base case, the EHMD was estimated to be cost-effective compared to the current nutrition for very low birth weight newborns with an incremental cost-effectiveness ratio (ICER) of €28,325 per Life-Year-Gained (LYG). From a societal perspective, the ICER is €27,494/LYG using a friction cost approach and €16,112/LYG using a human capital approach. Deterministic sensitivity analyses demonstrated that the estimate was robust against changes in the input parameters and probabilistic sensitivity analysis suggested that the probability EHMD was cost-effective at a threshold of €45,790/LYG was 94.8 percent.

is based was funded by an unrestricted research grant by Prolacta BioSciences (https://www.prolacta.com/). We acknowledge support for the Article Processing Charge by the Deutsche Forschungsgemeinschaft and the Open Access Publication Fund of Bielefeld University. The funder had no influece on the study design, data collection and analysis, decision to publish and the preparation of the manuscript. We acknowledge support for the Article Processing Charge by the Deutsche Forschungsgemeinschaft and the Open Access Publication Fund of Bielefeld University.

**Competing interests:** Both authors, SMS and WG, wish to disclose that the research on which our manuscript is based was funded by an unrestricted research grant by Prolacta BioSciences (https://www.prolacta.com/). The presence of the funding from a commercial source does not alter our adherence to PLOS ONE policies on sharing data and materials.

## Conclusion

Adopting EHMD as the standard approach to nutrition is a cost-effective intervention for very low birth weight newborns in Germany.

## Introduction

New-borns with very low birth weight show an increased risk of necrotising enterocolitis (NEC) [1], systemic sepsis [2, 3] and other clinical complications that may lead to infant mortality, short-term complications or even persistent, life-long sequelae. These include broncho-pulmonary dysplasia (BPD) [4, 5], retinopathy of prematurity (RoP) and neurodevelopmental problems [6]. While time trends suggest some improvements in the rates of sepsis and RoP, NEC rates remain constant and BPD rates have slightly increased [7]. Additionally the complications following a very low birth weight birth constitute 1.4 percent of all new-borns, corresponding to 11,051 very low birth weight babies in 2017 in Germany [8]. This number has remained relatively constant over time.

The feeding of very low birth weight babies seems to play a major role in the risk of developing complications and mother's milk seems to have a protecting effect against these complications compared to infant formula [9–14]. However, very low birth weight babies have a substantially greater requirement for a number of nutrients compared to full term babies. This larger demand is not met by mother's own milk or milk from donor milk banks [15], making the use of fortifiers necessary as a supplement to the milk [16, 17]. Human milk fortifiers and preterm formulas routinely used in neonatal feeding are derived from cow's milk, but recently fortifiers produced from donor mother's milk have become available. The feeding of very low birth weight babies with this exclusive human milk diet (EHMD) has shown a reduction in NEC, sepsis, BPD, RoP and mortality compared to cow milk-based nutrition [18–21].

In addition to the significant complications of a very low birth weight birth, the health care utilization of those babies also comes with high economic costs. Taking newborns with a birth weight of 750-999g as an example, the immediate inpatient care in neonatal intensive care units or regular care units including no significant operating procedure has a mean cost of €72,702 in Germany and increases to €121,594 for a birth weight below 600g [22]. The long-term effects were found to be €120.60 per patient per year for direct health care costs [23]. Reducing the complication rates associated with a very low birth weight birth may also lead to a reduction in the expenditure for neonatal and long-term care of these infants.

The aim of the current paper is to estimate the cost-effectiveness of an EHMD compared to the current nutritional practice for very low birth weight infants in Germany, taking into account short-term and long-term outcomes, as measured by life-years gained, and costs.

## Methods

### Model layout & natural history

The model has been constructed as a decision tree as depicted in Fig 1. For an EHMD or the current feeding strategy, very low birth rate neonates have a probability of developing NEC (ICD-10 P77) [24], sepsis (ICD-10 P36) [25], NEC and sepsis or no complication at all and may either survive or die from these health states or for other reasons [26]. NEC may be treated medically or surgically, and the latter may lead to short-bowel syndrome (SBS; ICD-10 K91.2) [27]. All such surviving newborns are additionally at risk of developing retinopathy of

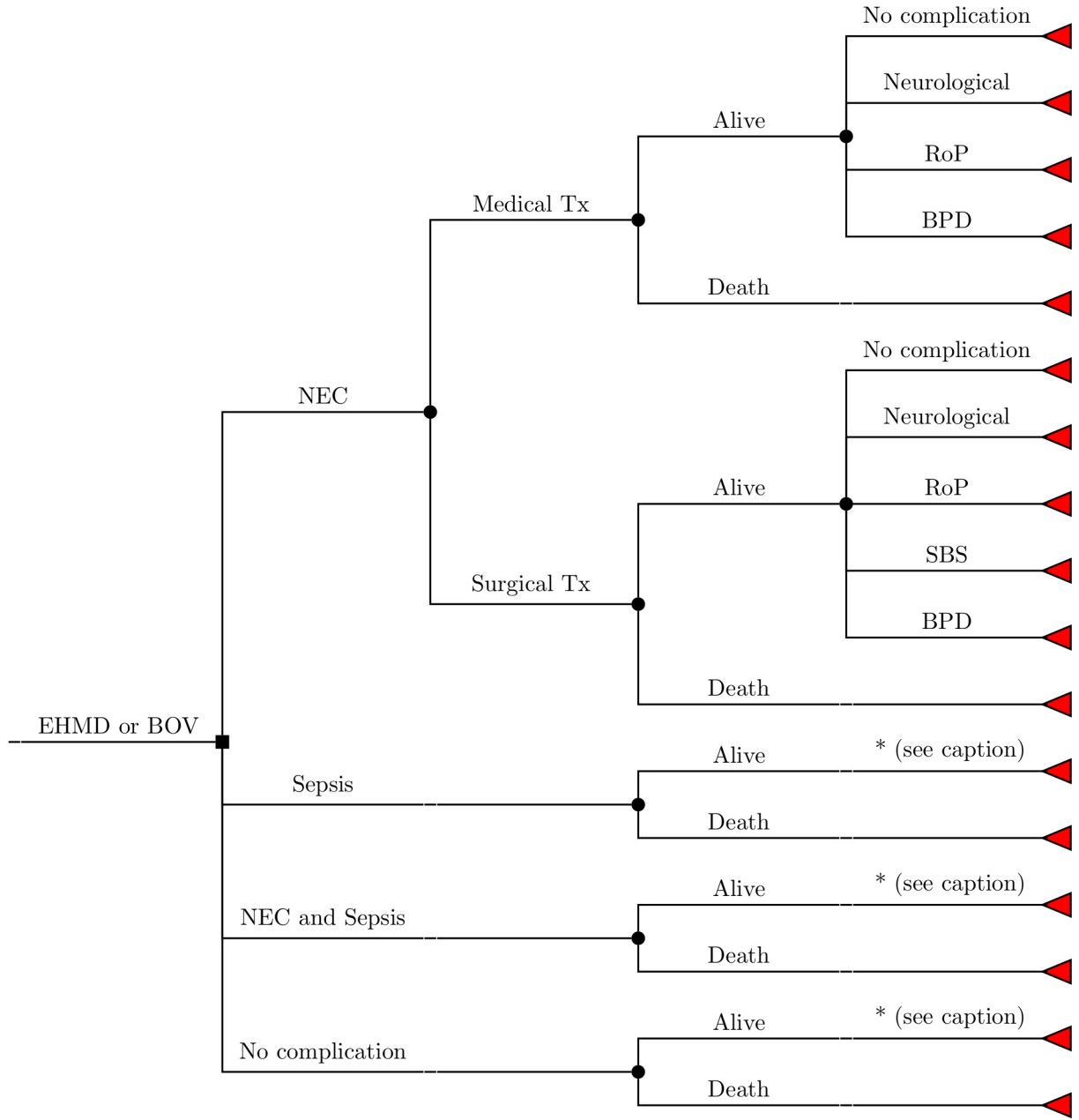

**Fig 1. Decision tree of the model.** For a better overview of the tree, the following simplifications were made in the above figure: Branches marked with an asterisk (*) follow the same branch as from the state "NEC medical". Death branches include health-state specific mortality as well as background mortality. (EHMD: exclusive human milk diet, BOV: cow milk-based nutrition, NEC: necrotising enterocolitis, Tx: treatment, RoP: retinopathy of prematurity, BPD: bronchopulmonary dysplasia, SBS: short bowel syndrome).

prematurity (RoP; ICD-10 H35.1) [26], bronchopulmonary dysplasia (BPD; ICD-10 P27.1) [28], cerebral palsy (ICD-10 G80) [6, 29] or other neurological impairments [6]. As the risks of these complications are dependent on the birth weight of the infant, the cohort running through the model has been stratified by birth weight: "<500g", "500-749g", "750-999g",

"1000-1249g" and "1250 -1499g". The time-horizon is life-long as all costs and outcomes are followed until the death of the cohort entering the model.

The model population consists of the average, yearly number of very low birth weight new-borns in Germany from the years 2012 to 2016 [30–34]. The probabilities of the different health states and complications are given in Table 1 for the overall cohort for the standard nutritional strategy. The birth weight-specific values can be found in the supplement. Wherever possible, health state specific probabilities of complications were directly calculated from extracted, stratified case numbers reported in various studies. When this was not possible (i.e. for the probabilities of NEC alone, sepsis alone and RoP given NEC and sepsis, respectively), relative risks have been calculated from available studies [26, 27] and were applied to German datasets to retrieve health state specific complication probabilities. Clopper-Pearson intervals for confidence intervals (CIs) on proportions have been calculated for parameters from studies without a control group. For studies with control groups, the CIs were calculated using a normal approximation with unequal group sizes.

## Intervention

An EHMD is administered dependent on the birth weight of the infant following the feeding protocol of the Texas Children's Hospital (TCH) in the base case [37]. In general, all feeding schemes start with a trophic feed, followed by an incremental increase in the feeding volume measured in ml per kg per day and then fortification. Upon reaching a specific weight goal and/or gestational age the new-borns are assumed to receive full feeds of 150ml/kg/day. Subsequently the infant is weaned off the human milk-derived fortifier over a four day period. In the base case, Prolact+6 is mixed with mother's milk in a 30ml to 70ml ratio. In a lesser fortification case, Prolact+4 is used in a ratio of 20/80ml and in the higher case a ratio of 40/60ml is assumed with Prolact+8.

The efficacy of EHMD has been taken from three different studies showing a relative risk (RR) of 40.0% [95% CI 30.4%–52.8%] against NEC [20, 38], 45.2% [95% CI 18.1%–113.1%] against surgical NEC [20, 38], 68.0% [95% CI 58.1%–79.5%] against sepsis [20, 38], 84.9% [95% CI 77.2%–93.3%] against BPD [38] and 55.1% [95% CI 38.4%–78.9%] against RoP [21, 38]. Study results are not yet available on the effectiveness of an EHMD on neurological sequelae, as these can only be assessed later in life. To explore the possible impact of an EHMD on the neurological development of very low birth weight infants, we evaluated a scenario with a RR of 0.90 against cerebral palsy and other neurological impairments. For the EHMD strategy, these RRs are applied to the corresponding parametric values in Table 1. The study population in Hair et al. (2016) included only infants between 750g and 1,250g. A specific analysis was carried out in the sensitivity analyses by calculating the results only for these weight groups.

## Costs

The cost of an EHMD was dependent on the weight-specific feeding scheme and is calculated with a price of €6 per ml for the fortifier. In the base case analysis it was assumed that the fortifier is added to the milk produced by the mother. The overall EHMD costs amount to €6,812 per person over all weight groups, i.e. the cost are €13,015 for infants <500g, €11,895 for 500-749g, €9,106 for 750-999g, €5,208 for 1,000–1,249g and €3,246 for the birth weight group of 1,250-1499g. Using milk from a donor milk bank at a price of €65 per litre (range €60 to €90; personal communication with German milk bank in Leipzig) was explored as a scenario in the sensitivity analyses. For the standard care strategy, no additional costs were incorporated into the model, as they were already included in the diagnosis-related groups (DRGs). Two

**Table 1. Medical model parameters.**

| Parameter | Mean | Range DSA | Distribution for PSA | Source |
|---|---|---|---|---|
| Cohort size | 9,519 | 8,136–10,903 | | [30–34] |
| Probability NEC* | 5.7% | 4.9%–6.5% | Dirichlet (Gamma(1;543)) | [24] |
| - NEC surgical | 25.1% | 21.8%–28.7% | Beta (682.05;2030.95) | [25] |
| - - NEC surgical BPD | 7.2% | 6.7%–7.8% | Beta (689.33;8829.07) | [30–34] |
| - - NEC surgical SBS | 19.4% | 15.9%–23.3% | Beta (88.81;369.19) | [27] |
| - - NEC surgical RoP* | 9.0% | 7.4%–10.8% | Beta (1,178.93;11,967.07) | [26] |
| - - NEC surgical Cerebral Palsy | 23.8% | 17.7%–30.9% | Beta (40.76;130.24) | [6, 29] |
| - - NEC surgical Neurological | 39.3% | 21.5%–59.4% | Beta (10.61;16.39) | [6] |
| - NEC medical | 74.9% | 71.3%–78.2% | Complementary probability to NEC surgical | |
| - - NEC medical BPD, RoP* | | | - Same as NEC surgical - | |
| - - NEC medical Cerebral Palsy | 11.6% | 7.4%–17.1% | Beta (21.88;166.12) | [6, 29] |
| - - NEC medical Neurological | 23.1% | 11.1%–39.3% | Beta (8.77;29.23) | [6] |
| - NEC surgical and medical mortality | 10.9% | 6.1%–15.7% | Beta(118.68;972.32) | [2] |
| - Background mortality | | | - Same as No complication mortality - | |
| Sepsis* | 12.7% | 12.1%–13.2% | Dirichlet Gamma(1;1,206)) | [25] |
| - - Sepsis BPD | | | - Same as NEC surgical - | |
| - - Sepsis RoP* | 2.8% | 2.3%–3.4% | Beta (368.02;12,777.98) | [26] |
| - - Sepsis Cerebral Palsy | 19.5% | 6.7%–32.3% | Beta (51.56;212.44) | [6, 35] |
| - - Sepsis Neurological | 35.9% | 29.9%–42.3% | Beta (87.64;156.36) | [6] |
| - Sepsis mortality | 17.5% | 23.3%–23.6% | Beta(32.83;155.17) | [36] |
| - Background mortality | | | - Same as No complication mortality - | |
| Sepsis + NEC | 2.3% | 1.8%–2.8% | Dirichlet (Gamma(1;217)) | |
| - Sepsis + NEC surgical | | | - Same as NEC surgical - | |
| - - Sepsis + NEC surgical BPD, RoP, SBS | | | - Same as NEC surgical - | |
| - - Sepsis + NEC surgical Cerebral Palsy | 21.4% | 4.7%–50.8% | Beta (2.79;10.21) | [6] |
| - - Sepsis + NEC surgical Neurological | 50.0% | 23.0%–77.0% | Beta (6.50;6.50) | [6] |
| - Sepsis + NEC medical | | | - Same as NEC medical - | |
| - - Sepsis + NEC medical BPD, RoP | | | - Same as NEC medical - | |
| - - Sepsis + NEC medical Cerebral Palsy | 20.0% | 5.7%–43.7% | Beta (3.80;15.20) | [6] |
| - - Sepsis + NEC medical Neurological | 45.0% | 23.1%–68.5% | Beta (8.55;10.45) | [6] |
| - Sepsis + NEC surgical mortality | | | - Combined probabilities of Sepsis and NEC surgical - | |
| - Sepsis + NEC medical mortality | | | - Combined probabilities of Sepsis and NEC medical - | |
| - Background mortality | | | - Same as No complication mortality - | |
| No NEC or sepsis | 79.3% | 77.4%–81.1% | Dirichlet Gamma(1;7,553)) | |
| - - No NEC or sepsis BPD | | | - Same as NEC surgical - | |
| - - No NEC or sepsis RoP* | 2.8% | 2.3%–3.4% | Beta (368.02;12,777.98) | [26] |
| - - No NEC or sepsis Cerebral Palsy | 15.0% | 12.6%–17.6% | Beta (120.85;687.15) | [6] |
| - - No NEC or sepsis Neurological | 22.0% | 19.2%–25.0% | Beta (177.78;630.22) | [6] |
| - No complication mortality | 14.9% | 13.7%–16.2% | Beta (838.92;5,752.05) | [26] |

Parameters marked with a * are calculated using relative risks from other studies. (DSA: deterministic sensitivity analysis, PSA: probabilistic sensitivity analysis, NEC: necrotising enterocolitis, BPD: bronchopulmonary dysplasia, SBS: short bowel syndrome, RoP: retinopathy of prematurity)

additional feeding scenarios (using less and more fortifier, respectively) have been used as the lower and upper bound for the costs of an EHMD, but assuming the same degree of effectiveness.

The costs for each health state were calculated via a price-resources framework for inpatient stays. The length of stay (LOS) was been extracted for each ICD code from the database of the

Federal Statistical Office covering the period 2006 and 2016 [39] and the mean LOS was combined with the corresponding Diagnosis Related Groups (DRGs) [22] to calculate the inpatient costs using the base rate of €3,376 for the year 2017 [40]. As the ICD-specific LOS data stratified by birth weight were not available, adjustment factors have been calculated from another German data source on LOS for very low birth weight infants by calculating the ratio of the LOS of the different birth weight categories and the overall LOS of all newborns with a birth weight between 500g and 1499g [25]. Costs for complications (BPD, SBS, RoP) were calculated accordingly and added to the health state-specific costs. Lifetime, direct health care costs were retrieved from a German study [41] for cerebral palsy (€232,703 at 3% discounting) and for other neurological impairments (€94,220).

For the societal perspective, indirect costs were calculated using the human capital approach, including the forgone productivity due to death or disability measured by the discounted life-time earnings of a newborn and the productivity loss of one parent during the inpatient stay. A scenario of the friction cost approach has also been applied. Using this approach, one parent staying with the newborn has a productivity loss for the friction period of 92 days [42]. The reference year of all costs is 2017.

## Analysis

The base case analysis was conducted from a third-party payer (TPP) perspective with 3% discount rates on costs and LYG using the TCH (+6) feeding scheme for EHMD and the friction cost approach as recommended by German guidelines [43]. Discount rates of 0% and 5% as well as a societal perspective were also explored. In the absence of an official threshold of the maximal acceptable willingness-to pay for an additional LYG, the GDP per capita ($50,878/€45,790 [44, 45]) from the recommendations of the World Health Organization (WHO) was used [46]. Further deterministic sensitivity analyses (DSA) were conducted by setting parameters at their lower and upper bound, which were set from the 95% CIs wherever possible. Break-even analyses have been performed on the probabilities for NEC and sepsis for a threshold of €50,000/LYG in order to explore the robustness of the cost-effectiveness calculation. A probabilistic sensitivity analysis (PSA) was conducted using the probability distributions given in Table 1 and the appendix, drawing 1,000 parametric values from those distributions. The probabilities of the intervention being cost-effective were calculated for willingness-to-pay (WTP) thresholds from €0 to €100,000 per LYG and have been plotted as cost-effectiveness acceptability curves (CEACs). Additionally, an expected value of perfect information (EVPI) analysis was conducted to estimate the cost-effectiveness of further research. Internal validity of the model was checked by calculating the results of the total population once directly and one via weighted averages over the different birth weight strata. All analyses were conducted in Microsoft Excel 2013.

## Results

In the base case analysis, assuming the Texas Children's Hospital+6 feeding strategy, a third party payer (TPP) perspective and no use of donor milk, the cost-effectiveness of EHMD is €35,464 per LYG. Using the friction cost approach, the corresponding value for the societal perspective is €33,991/LYG. The cost-effectiveness from a societal perspective using the human capital approach improves to €9,681/LYG. The sub-group analysis for the different birth weight groups reveals an association between lower birth weight and a more favourable ICER. The ICER is €16,272/LYG in the base case for infants in the "<500g" group and increases to €70,910/LYG in the category "1250-1499g". The cost-effectiveness for the clinical study population based on an EHMD (750g to 1249g) is €34,016/LYG from a TPP perspective

**Table 2. Direct and indirect costs and life years lost for the different birth-weight groups of the study population for the base case.**

| Birth weight | N | Direct Costs per patient | | Indirect Costs per patient | | Life years lost | | ICER | |
|---|---|---|---|---|---|---|---|---|---|
| | | Status quo | EHMD | Status quo | EHMD | Status quo | EHMD | Societal | TPP |
| <500g | 475 | €64,148 | €91,242 | €4,345 | €2,715 | 20.91 | 19.24 | €15,293 | €16,272 |
| 500-749g | 1,540 | €87,103 | €112,021 | €7,054 | €4,608 | 13.88 | 12.72 | €19,274 | €21,371 |
| 750-999g | 2,004 | €97,283 | €121,688 | €8,645 | €6,368 | 6.89 | 5.91 | €22,579 | €24,882 |
| 1000-1249g | 2,166 | €64,585 | €84,083 | €3,838 | €3,573 | 2.45 | 2.14 | €62,076 | €62,931 |
| 1250-1499g | 3,335 | €58,736 | €76,596 | €3,487 | €3,863 | 1.40 | 1.15 | €72,402 | €70,910 |
| Total | 9,519 | €73,041 | €94,254 | €5,273 | €4,391 | 5.79 | 5.19 | €33,991 | €35,464 |

(ICER: incremental cost-effectiveness ratio, EHMD: exclusive human milk diet, TPP: third-party payer)

and €32,061/LYG or €8,183/LYG from a societal perspective using the friction cost or the human capital approach. Detailed results for the base case can be found in Table 2.

The deterministic sensitivity analyses (i.e. setting input parameter values to their lower and upper bounds) reveal the probability for death due to sepsis and the efficacy with regards to mortality reduction to be the most influential parameters on the ICER value. Setting the sepsis mortality to the lower bound of 12.3% yields an ICER of €43,703/LYG, and €29,043/LYG for the upper bound of 23.6%. However, changing the RR for the health state-specific mortality yields ICERs of €44,902 and €30,584/LYG for the lower (0.90) and upper bounds (0.57), respectively. As can be seen in Fig 2, the probabilities of the health states and the effectiveness of an EHMD are found among the most influential parameters. Assuming different feeding strategies has only a minor effect on the ICER (see Price EHMD). The conservative feeding strategy results in an ICER of €31,548/LYG and the higher dosage of the fortifier results in an ICER of €36.612/LYG. No single parameter leads to the ICER being above the WTP threshold of €45,790/LYG.

In the scenario analyses, further changes to the model assumptions have been explored. Firstly, assuming the use of donor milk instead of mothers own milk for an EHMD, slightly increases the ICER to €35,867/LYG from a TPP perspective and to €34,394/LYG from a societal perspective. Secondly, the costs of the formula in the current feeding have explicitly been included in the model to reflect the replacement of currently used formula by an EHMD. Including costs of the current formula at a price of €20 per 200g (overall €223 per infant), the ICER would decrease to €35,091/LYG from a TPP perspective and to €33,618/LYG from a societal perspective with the friction approach. If no protective effect of an EHMD against neurological impairments and cerebral palsy was assumed, the cost-effectiveness increased to €43,459/LYG from a TPP perspective (€42,110/LYG from a societal perspective).

Furthermore, several break-even analyses were conducted to evaluate the influence of key parameters beyond their statistical uncertainty had on the ICER. As the probability of NEC was from a non-German source, the NEC probability has been lowered below the lower 95% CI of 4.9 percent down to zero. The ICER for this scenario was €52,762/LYG, while increasing the probability of NEC leads to improvements of the base case ICER. The probability of sepsis would need to be set to 5.5 percent (lower 95% CI in the base case: 12.1 percent) for the ICER to reach the WTP threshold of €45,790. The costs of an EHMD could be increased by approximately 25% (i.e. €8,516 per new-born) above the base case costs before the cost-effectiveness would be above the WTP threshold.

The result of the PSA for 1,000 model runs is shown in Fig 3 in the form of a CEAC. Until a WTP threshold of €25,000/LYG, the probability for an EHMD being cost-effective is close to zero. After this threshold, the probability of an EHMD being cost-effective shows a steep

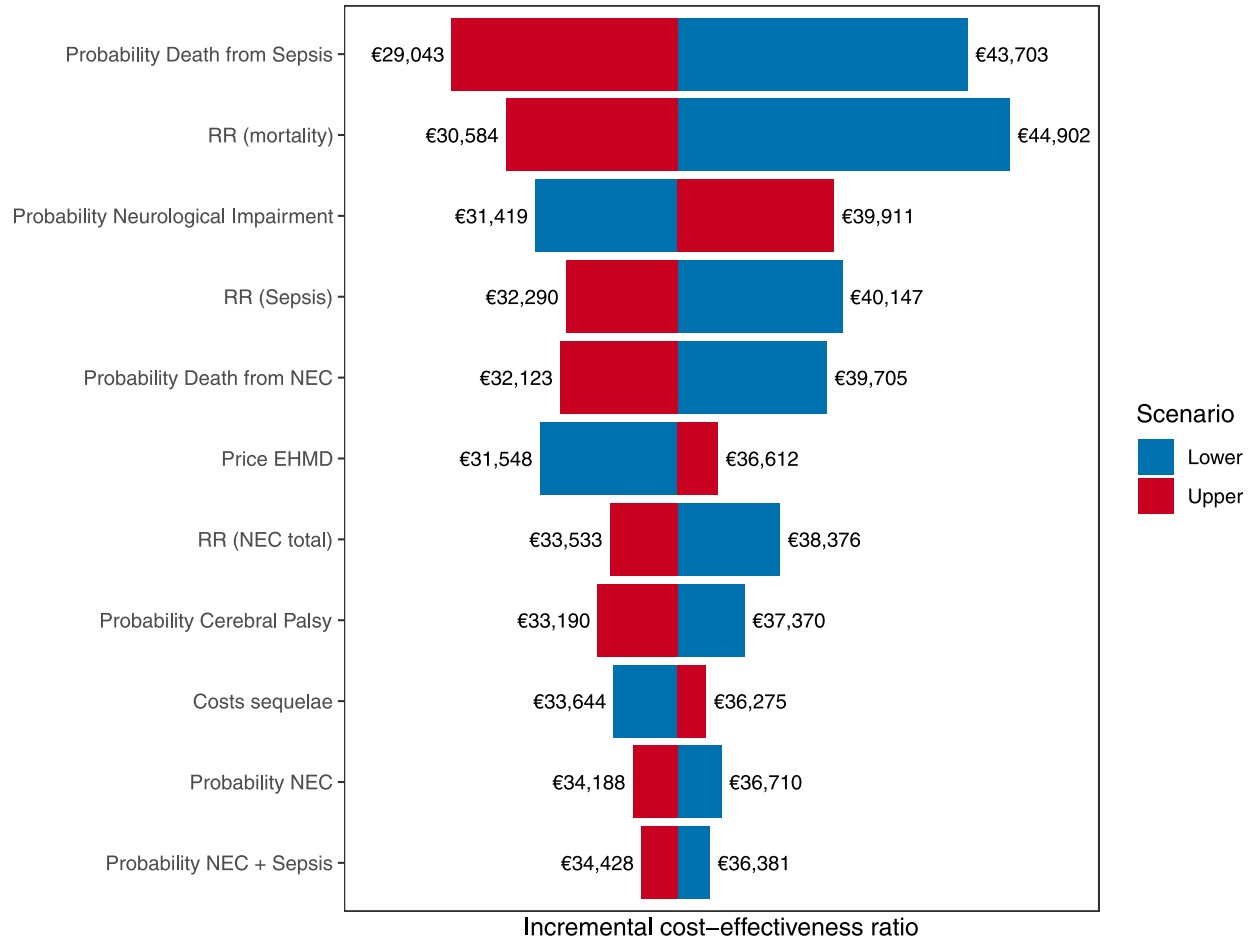

**Fig 2. Tornado plot for all input parameters with an influence of more than €1,000 on the incremental cost-effectiveness ratio (ICER).** (RR: relative risk, NEC: necrotising enterocolitis, EHMD: exclusive human milk diet).

incline, until the probability of EHMD being cost-effective reaches 97 percent at a threshold of €60,000/LYG. Eighty-one percent of the 1,000 model runs show an EHMD to be cost-effective at the WTP threshold of €45,790/LYG.

Based on these model iterations of the PSA, the population EVPI has been calculated and the results are shown in Fig 4. At the maximal WTP threshold of €45,790/LYG, the population EVPI corresponds to €5,610,000, or €589 per new-born per year. Conducting future research for this amount may potentially be cost-effective if the WTP threshold is above €31,000/LYG and below or equal the maximal WTP threshold.

## Discussion

The results of this modelling study need to be interpreted in the light of several limitations. First of all, the study population of the clinical studies used for the EHMD efficacy is from the US and does not entirely match the German model population in terms of clinical complications and birth weight. The results for the birth weight subgroups from the clinical studies was very similar to the overall cost-effectiveness (€35,464 vs. €34,016 per LYG, respectively), but it remains unclear if the effectiveness estimates can be transferred to infants with birth weights

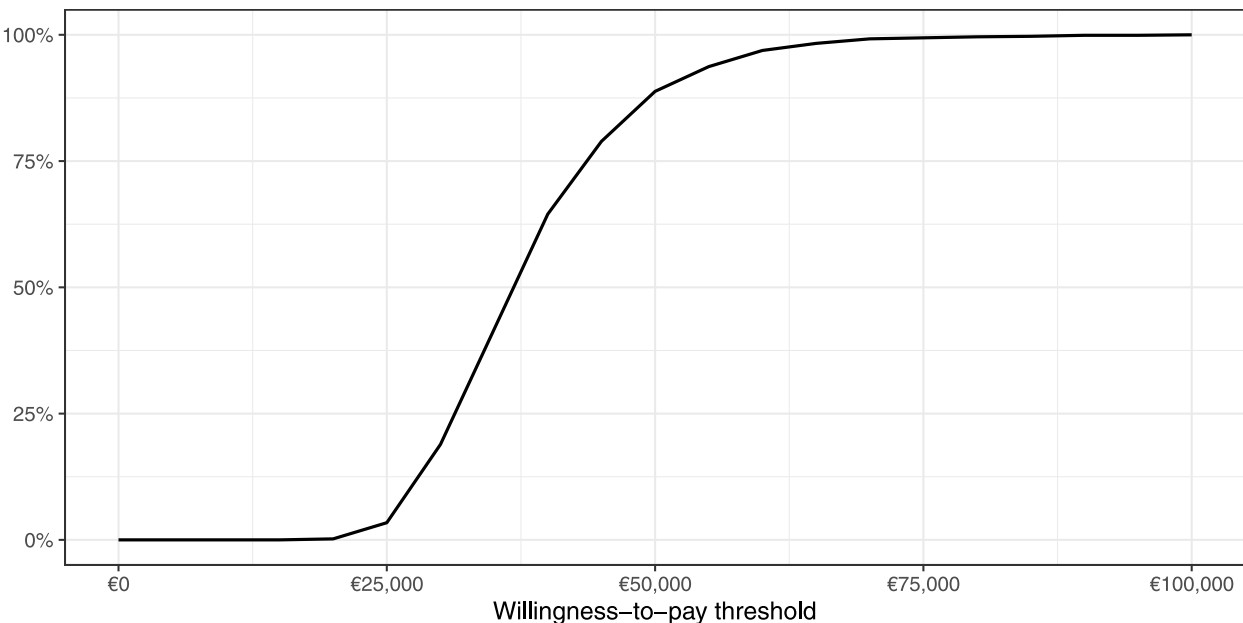

**Fig 3. Cost-effectiveness acceptability curve for 1,000 model iterations.**

below 750g or above 1,250g. The transferability of our results is also limited with regard to other countries due to the specific NEC rates, the German reimbursement scheme for inpatient costs and the cost of human milk and fortifiers in Germany. Secondly, the control group in the clinical studies received mother's milk with a bovine milk-based fortifier. As feeding protocols may vary between hospitals, it is unclear if all very low birth weight infants in

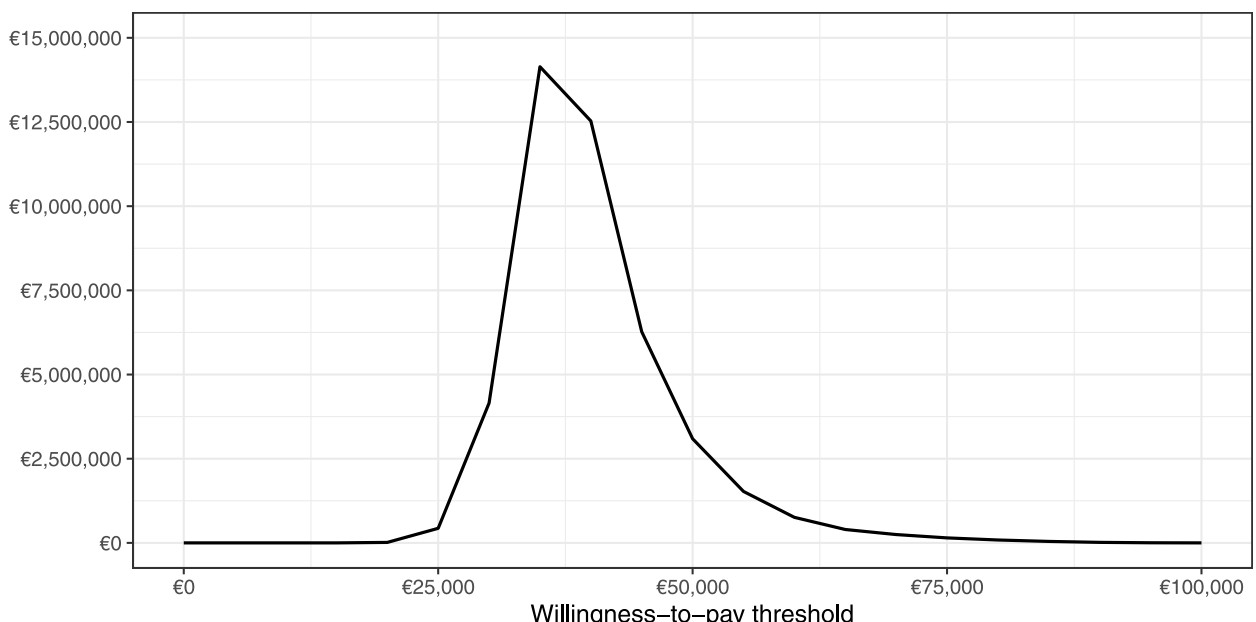

**Fig 4. Expected value of perfect information for the population of 9,519 very low birth weight new-borns per year in Germany.**

Germany currently receive the exact same feeding as the control group of the studies considered. However, the current guidelines only mention bovine-based fortifiers [16] and it can be reasonably assumed that the infants represented by the German data used in the model were not fed using an EHMD. In this context it should also be stated that the effectiveness of an EHMD was only considered against the health state specific mortality. If a very low birth weight infant dies, there may be several conditions present that may have caused the death, making it hard to calculate health-state specific mortality rates. Thus, there may be deaths in the model from background mortality that may also be prevented by an EHMD. Additionally, the mortality of newborns suffering from concurrent NEC and sepsis in the model is assumed to be the statistically independent combination of the respective NEC and sepsis mortalities when it may in fact be higher. This would lead to an underestimation of the effectiveness of an EHMD.

With regard to the main outcome of the model, LYG does not capture all consequences of a disease. Quality adjusted life-years (QALYs) are usually taken as the primary outcome in economic evaluation to cover not only mortality but also the morbidity associated with a disease. Unfortunately, the elicitation of utilities for the different health states in newborns is not easily possible and QALYs are, therefore, not available for the evaluation of interventions for these infants. This also implies that spill-over [47] and bereavement effects [48], i.e. a quality of life loss for parents by the disability or the death of their child, cannot be incorporated in the model.

The economic results of the model need also to be seen with respect to the medical outcomes of the model. While the EHMD strategy avoids around 290 deaths, there are more survivors who may suffer from long-term complications. While there are fewer cases of BPD, SBS and RoP for the EHMD strategy, there are 30 more cases of cerebral palsy and 25 more cases of other neurological impairments compared to the control. Further research may focus on the long-term effects of human milk-based fortifiers on the neurological development of these infants. The EVPI analysis revealed that this future research might be cost-effective under the current WTP threshold of up to €6 million per year.

## Conclusions

The EHMD strategy can be considered a cost-effective new treatment strategy for very low birth weight newborns in Germany from a TPP perspective under a maximal WTP threshold of €45,790/LYG. The deterministic as well as the probabilistic sensitivity analyses showed that this estimate is robust against a variety of changes to the input parameters values. Only decreasing the effectiveness against more than one complication concurrently makes the ICER increase above the WTP threshold of €45,790/LYG recommended by WHO for Germany.

## Supporting information

**S1 File. Supporting information.** This file contains a detailed list of all model parameters. (PDF)

## Acknowledgments

We acknowledge support for the Article Processing Charge by the Deutsche Forschungsgemeinschaft and the Open Access Publication Fund of Bielefeld University.

## Author Contributions

**Conceptualization:** Wolfgang Greiner.

**Formal analysis:** Stefan Michael Scholz.

**Funding acquisition:** Wolfgang Greiner.

**Methodology:** Stefan Michael Scholz.

**Project administration:** Wolfgang Greiner.

**Software:** Stefan Michael Scholz.

**Supervision:** Stefan Michael Scholz.

**Validation:** Stefan Michael Scholz.

**Visualization:** Stefan Michael Scholz.

**Writing – original draft:** Stefan Michael Scholz.

**Writing – review & editing:** Wolfgang Greiner.

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
