## [Decision Letter · Decision Letter 0]

3 Oct 2019

PONE-D-19-22800

An Exclusive Human Milk Diet for Very Low Birth Weight New-borns – a Cost-Effectiveness and EVPI Study for Germany

PLOS ONE

Dear Mr. Scholz,

Thank you for submitting your manuscript to PLOS ONE. After careful consideration, we feel that it has merit but does not fully meet PLOS ONE’s publication criteria as it currently stands. Therefore, we invite you to submit a revised version of the manuscript that addresses the points raised during the review process.

We would appreciate receiving your revised manuscript by Nov 17 2019 11:59PM. To enhance the reproducibility of your results, we recommend that if applicable you deposit your laboratory protocols in protocols.io, where a protocol can be assigned its own identifier (DOI) such that it can be cited independently in the future. For instructions see: http://journals.plos.org/plosone/s/submission-guidelines#loc-laboratory-protocols

We look forward to receiving your revised manuscript.

Kind regards,

Prem Shekhawat, MD

Academic Editor

PLOS ONE

Journal Requirements:

1.  Thank you for your funding statement; "Both authors, SMS and WG, wish to disclose that the research on which our manuscript is based was funded by an unrestricted research grant by Prolacta BioSciences (https://www.prolacta.com/). The funder had no influece on the study design, data collection and analysis, decision to publish and the preparation of the manuscript. "

We note that you received funding from a commercial source: Prolacta BioSciences

Additional Editor Comments (if provided):

Thank you for your submission titled "An Exclusive Human Milk Diet for Very Low Birth Weight New-borns – a Cost-Effectiveness and EVPI Study for Germany", it made interesting reading but needs major revision before it can be considered suitable for publication. The issues which must be addressed are:

1. Major English language corrections are required, there are too many problems which need fixing before it can be considered suitable for further review. I suggest hiring a language expert who should essentially rewrite the whole manuscript so that it becomes easy to read so that average reader can understand the meaning.

2. Manuscript is statistics heavy, tables are too long, results presented in a very complex manner which are hard to understand for average reader of PLOS one, kindly shorten data, every single detail need not be provided but salient features are shown in tables, cutting back on some figures and tables will help.

3. It should be very clear in results and discussion section that this study results are valid only to the local German population and cannot be applied universally since the NEC rate as well as cost of human milk fortifier are highly variable between centers. Likewise hospital charges are much different then actual cost of the product making this work difficult to use.

Kindly see attached comments by two experts in the field to revise your manuscript.

Reviewers' comments:

Reviewer's Responses to Questions

**Comments to the Author**

1. Is the manuscript technically sound, and do the data support the conclusions?

Reviewer #1: Yes

Reviewer #2: Yes

2. Has the statistical analysis been performed appropriately and rigorously? 

Reviewer #1: I Don't Know

Reviewer #2: Yes

3. Have the authors made all data underlying the findings in their manuscript fully available?

Reviewer #1: Yes

Reviewer #2: Yes

4. Is the manuscript presented in an intelligible fashion and written in standard English?

Reviewer #1: Yes

Reviewer #2: Yes

5. Review Comments to the Author

Reviewer #1: Line 34 : Please provide the expansion of LYG and ICER here

Line 48 : need to be BPD instead of BDP

Line 108-109 : what is meant by the statement"in cost scenario for EHMD a ratio of 30/70 is assumed using Prolact+8 (with a ratio of 40/60)". Why is it assumed as 30/70 when its 40/60

Line 115-116 : If there are no available data for the effectiveness of EHMD on neurological sequelae, based on what criteria was the RR of 0.90 assumed

Line 136-138 : What was the process used to calculate the LOS data based on birth weight? What is the adjustment factor and how was it arrived at?

Line 160 : Put WTP in parenthesis here so that the reader can recognize what it stands for later in the manuscript.

Line 177 - its showing both Table 2Table 1. which one is it? Table 2?

Table 2 : Please add "for the base case" in the legend.

Reviewer #2: The feeding of very low birth weight babies seems to play a major role in the risk of developing complications. Scholz and Greiner implemented a decision-tree approach to investigate the cost-effectiveness between human milk-based fortifiers and cos’s milk-based fortifiers in very low birth weight new-borns. They concluded that an exclusive human milk diet is a cost-effective intervention.

1. Table or Figure should be self-explained. The abbreviation should be noted, e.g. what are DSA, PSA and so on.

2. Authors mentioned that the health state specific probabilities of complication were either directly calculated or estimated from other studies. It would be good to indicate which is from direct calculation and which is estimated from other studies in the presentation of results/tables.

3. Some estimates, e.g. health state specific probabilities, the efficacy of EHMD and so on, were not directly calculated based on the sample but estimated from other studies. For those from other studies, please comment on how compatible or generalizable to the German population.

6. PLOS authors have the option to publish the peer review history of their article (what does this mean?). If published, this will include your full peer review and any attached files.

Reviewer #1: No

Reviewer #2: No

---

## [Author Response · Author response to Decision Letter 0]

4 Nov 2019

Reviewer #1: 

Thank you very much for taking the time to review our paper and thank you for your constructive feedback. We are grateful you even specified the lines where changes need to be made. Please see below our responses on your comments, which where all included in the new version of the manuscript.

Line 34 : Please provide the expansion of LYG and ICER here

Thank you very much, we added the expansion of LYG and ICER.

Line 48 : need to be BPD instead of BDP

Thank you for noticing this. The typo has been corrected.

Line 108-109 : what is meant by the statement"in cost scenario for EHMD a ratio of 30/70 is assumed using Prolact+8 (with a ratio of 40/60)". Why is it assumed as 30/70 when its 40/60

Sorry for this relict of a previous version. The correct ratio is 40/60 and the sentence has been changed accordingly. It now reads as follows: 

“In the lower scenario, Prolact+4 is used with a ratio of 20/80ml and in the high cost scenario for EHMD a ratio of 40/60 is assumed using Prolact+8.”

Line 115-116 : If there are no available data for the effectiveness of EHMD on neurological sequelae, based on what criteria was the RR of 0.90 assumed

There is no direct evidence for the effectiveness of EHMD on neurological sequelae as the manifestation of a neurological impairment will only become evident when the children are at around school age. But there are studies suggesting that NEC and sepsis are associated with a higher probability of neurological impairment and reducing NEC and sepsis thus also leads to a reduction in neurological impairments (please see references below). We therefore included – from our perspective – a conservative value of 0.90 for neurological sequelae. Please let us know if this acceptable in your opinion.

Roze et al. 2011 Functional Impairments at School Age of Children With Necrotizing Enterocolitis or Spontaneous Intestinal Perforation, Pediatric Research 70(6):619-625.

van der Ree et al. 2011, Functional impairments at school age of preterm born children with late-onset sepsis, Early Human Development 87:821-826.

Ehrenkranz et al. 2006, Growth in the Neonatal Intensive Care Unit Influences Neurodevelopmental and Growth Outcomes of Extremely Low Birth Weight Infants, Pediatrics 117(4):1253-1261.

Line 136-138 : What was the process used to calculate the LOS data based on birth weight? What is the adjustment factor and how was it arrived at?

Unfortunately, there is no data source available that would allow estimating the LOS for the different conditions (NEC, sepsis, etc.) stratified by age. However, there are two different data sources available that give information about the marginal distributions of the LOS by birth weight and the LOS for each of the conditions, respectively. We used the data source giving the LOS by birth weight to calculate the ratio of the LOS for each birth weight category, compared to the LOS for all new-borns with a birth weight between 500g and 1499g.The LOS for the different conditions were then multiplied by this factor. We added some more information on the calculation and the sentence now reads as follows: 

“As the ICD-specific LOS data were not available stratified by birth weight, adjustment factors have been calculated from another German data source on LOS of very low birth weight infants by calculating the ratio of the LOS of the different birth weight categories and the overall LOS of all new-borns with a birth weight between 500g and 1499g.”

Line 160 : Put WTP in parenthesis here so that the reader can recognize what it stands for later in the manuscript.

The abbreviation WTP for willingness-to-pay has been added.

Line 177 - its showing both Table 2Table 1. which one is it? Table 2?

Sorry, this seems to be a problem with the automated numbering in Word. We removed all automated fields. It should read as Table 2.

Table 2 : Please add "for the base case" in the legend.

We amended the table caption as you suggested.

Reviewer #2: 

1. Table or Figure should be self-explained. The abbreviation should be noted, e.g. what are DSA, PSA and so on.

Thank you for this suggestion. We added the explanations of the abbreviations used in the table and figures in the respective captions.

2. Authors mentioned that the health state specific probabilities of complication were either directly calculated or estimated from other studies. It would be good to indicate which is from direct calculation and which is estimated from other studies in the presentation of results/tables.

Thank you for this comment. We included a sentence in the methods section that states the probabilities which were estimated from other studies. The sentence now reads: “When this was not possible (i.e. for the probabilities of NEC alone and RoP given NEC and sepsis, respectively), relative risks have been calculated from studies [26, 27] and were applied to German datasets to retrieve health state specific complication probabilities.” We also added asterisks to parameters in table 1 for which other studies were used.

3. Some estimates, e.g. health state specific probabilities, the efficacy of EHMD and so on, were not directly calculated based on the sample but estimated from other studies. For those from other studies, please comment on how compatible or generalizable to the German population.

You are correct, that the transferability of the results from the other studies is limited. We amended the limitations section in the discussion to put more emphasis on this point. The discussion starts as follows:

“The results of this modelling study need to be interpreted in the light of several limitations. First of all, the study population of the clinical studies on EHMD efficacy is from the US and does not entirely match the German model population by the clinical complications and birth weight. The results for the sub-groups of the model population corresponding to the birth weights of the clinical studies were very similar to the overall cost-effectiveness (€35,464 vs. €34,016 per LYG, respectively), but it remains unclear if the effectiveness estimates can be transferred to new-borns with birth weights below 750g or above 1,250g. The transferability of our results is also limited with regard to other countries due to the specific NEC rates, the German reimbursement scheme for inpatient costs and the cost of human milk and fortifiers in Germany.“

---

## [Decision Letter · Decision Letter 1]

2 Dec 2019

An Exclusive Human Milk Diet for Very Low Birth Weight Newborns – a Cost-Effectiveness and EVPI Study for Germany

PONE-D-19-22800R1

Dear Dr. Scholz,

We are pleased to inform you that your manuscript has been judged scientifically suitable for publication and will be formally accepted for publication once it complies with all outstanding technical requirements.

With kind regards,

Prem Singh Shekhawat, MD

Academic Editor

PLOS ONE

Additional Editor Comments (optional):

Reviewers' comments:

Reviewer's Responses to Questions

**Comments to the Author**

1. If the authors have adequately addressed your comments raised in a previous round of review and you feel that this manuscript is now acceptable for publication, you may indicate that here to bypass the “Comments to the Author” section, enter your conflict of interest statement in the “Confidential to Editor” section, and submit your "Accept" recommendation.

Reviewer #1: All comments have been addressed

Reviewer #2: All comments have been addressed

2. Is the manuscript technically sound, and do the data support the conclusions?

Reviewer #1: (No Response)

Reviewer #2: (No Response)

3. Has the statistical analysis been performed appropriately and rigorously? 

Reviewer #1: (No Response)

Reviewer #2: (No Response)

4. Have the authors made all data underlying the findings in their manuscript fully available?

Reviewer #1: (No Response)

Reviewer #2: (No Response)

5. Is the manuscript presented in an intelligible fashion and written in standard English?

Reviewer #1: (No Response)

Reviewer #2: (No Response)

6. Review Comments to the Author

Reviewer #1: (No Response)

Reviewer #2: (No Response)

7. PLOS authors have the option to publish the peer review history of their article (what does this mean?). If published, this will include your full peer review and any attached files.

Reviewer #1: No

Reviewer #2: No

---

## [Editor Report · Acceptance letter]

16 Dec 2019

PONE-D-19-22800R1 

An Exclusive Human Milk Diet for Very Low Birth Weight Newborns – a Cost-Effectiveness and EVPI Study for Germany 

Dear Dr. Scholz:

I am pleased to inform you that your manuscript has been deemed suitable for publication in PLOS ONE. Congratulations! Your manuscript is now with our production department. 

With kind regards,

on behalf of

Dr. Prem Singh Shekhawat 

Academic Editor

PLOS ONE